# Differentiable Rendering with Reparameterized Volume Sampling

## Abstract

We propose an alternative rendering algorithm for neural radiance fields based on importance sampling. In view synthesis, a neural radiance field approximates underlying density and radiance fields based on a sparse set of scene views. To generate a pixel of a novel view, it marches a ray through the pixel and computes a weighted sum of radiance emitted from a dense set of ray points. This rendering algorithm is fully differentiable and facilitates gradient-based optimization of the fields. However, in practice, only a tiny opaque portion of the ray contributes most of the radiance to the sum. Therefore, we can avoid computing radiance in the rest part. In this work, we use importance sampling to pick non-transparent points on the ray. Specifically, we generate samples according to the probability distribution induced by the density field. Our main contribution is the reparameterization of the sampling algorithm. It allows end-to-end learning with gradient descent as in the original rendering algorithm. With our approach, we can optimize a neural radiance field with just a few radiance field evaluations per ray. As a result, we alleviate the costs associated with the color component of the neural radiance field at the additional cost of the density sampling algorithm.

## 1 Introduction

We propose a volume rendering algorithm for learning 3D scenes and generating novel views. Recently, learning-based approaches led to significant progress in this area. As an early instance, [8] proposed to represent a scene via a density field and a radiance (color) field parameterized with an MLP. They run a differentiable volume rendering algorithm with the MLP-based fields and minimize the discrepancy between the produced images and a set of reference images to learn a scene representation. The algorithm we propose is a drop-in replacement for the volume rendering algorithm used in NeRF [8] and follow-ups.

The underlying model in NeRF generates an image point in the following way. It casts a ray from a camera through the point and defines the point color as a weighted sum along the ray. The sum aggregates the radiance of each ray point with weights induced by the density field. Each summand involves a costly neural network query, and model has a trade-off between rendering quality and computational load. NeRF obtained a better trade-off with a two-stage sampling algorithm used to get ray points with higher weights. The algorithm is reminiscent of importance sampling, yet it requires training an auxiliary model.

In this work we propose a rendering algorithm based on importance sampling. Our algorithm also acts in two stages. In the first stage, it marches through the ray to estimate density. In the second stage, it constructs a Monte-Carlo color approximation using the density to pick points along the ray. The resulting estimate is fully-differentiable and does not require any auxiliary models. Besides that, we only need a few samples to construct precise color approximation. An intuitive explanation is that

37 we only need to compute the radiance of the point where a ray hits a solid surface. In the experiments,
38 we query radiance for $\times 16$ fewer ray points during training compared to baseline. Nevertheless, we
39 manage to obtain competitive model and rendering quality.

40 As a result, our algorithm is more suitable for recent solutions [10, 13, 12] that use distinct models to
41 parameterize radiance and density. Specifically, the first stage only queries the density field, whereas
42 the second stage only queries the radiance field. Compared to the standard rendering algorithm, the
43 second stage of our algorithm avoids redundant radiance queries and reduces the memory required
44 for rendering.

## 2   Neural Radiance Fields

46 Neural radiance fields represent 3D scenes with a scalar density field $\sigma : \mathbb{R}^3 \to \mathbb{R}^+$ and a vector
47 radiance field $c : \mathbb{R}^3 \times R^3 \to \mathbb{R}^3$. The scalar field $\sigma$ represents volume density at each spatial
48 location $\boldsymbol{x}$, and $c(\boldsymbol{x}, \boldsymbol{d})$ returns the light emitted from spatial location $x$ in directionn $d$ represented as
49 a normalized three dimensional vector.

50 For novel view synthesis, they adapt a volume rendering technique that computes a pixel color
51 $C(\boldsymbol{o}, \boldsymbol{d})$ (denoted with a capital letter). In particular, the expected color along a ray $\boldsymbol{r} = \boldsymbol{o} + t\boldsymbol{d}$ going
52 from the camera through the pixel is

$$C(\boldsymbol{o}, \boldsymbol{d}) = \int_{t_n}^{+\infty} p_{\boldsymbol{r}}(t) c(\boldsymbol{o} + t\boldsymbol{d}, \boldsymbol{d}) \mathrm{dt}, \text{ for } p_{\boldsymbol{r}}(t) = \sigma(\boldsymbol{o} + t\boldsymbol{d}) \exp\left(-\int_{t_n}^{t} \sigma(\boldsymbol{o} + s\boldsymbol{d}) \mathrm{ds}\right). \quad (1)$$

53 Here, $p_{\boldsymbol{r}}(t)$ is a probability density function of a random variable $T$ on a ray $r$. Intuitively, $T$ is the
54 location on the ray where a portion of light coming into the point $\boldsymbol{o}$ was emitted.

55 One way to approximate to the integral would be to cut off the integral at depth $t_f$ and then use a grid
56 $t_n = t_0 < t_1 < \cdots < t_m = t_f$ to compute the integral with a Riemann sum

$$\hat{C}_{\text{Riemann}}(\boldsymbol{o}, \boldsymbol{d}) = \sum_{i=1}^{m}(t_i - t_{i-1}) p_{\boldsymbol{r},i} c(\boldsymbol{o} + t_i \boldsymbol{d}, \boldsymbol{d}), \quad (2)$$

$$\text{where } p_{\boldsymbol{r},i} = \sigma(\boldsymbol{o} + t_i \boldsymbol{d}) \exp\left(-\sum_{j=1}^{i}(t_j - t_{j-1})\sigma(\boldsymbol{o} + t_j \boldsymbol{d})\right). \quad (3)$$

57 Importantly, Eq 2 is fully differentiable and can be used as a part of gradient-based learning pipeline.

58 While such approximation works in practice, a fauithfull approximation requires a dense grid and
59 multiple evaluations of $\sigma$ and $c$. Besides that, a common situation is when a ray intesects a solid
60 surface at some point $s \in [t_n, t_f]$. In this case, probability density $p_{\boldsymbol{r}}(t)$ will concentrate its mass
61 near $s$ and will be close to zero in other parts of the ray. As a result, most of the summands in Eq. 2
62 will make negligible contribution to the sum.

63 Monte Carlo methods give another way to apporximate the color. Given $n$ i.i.d. samples $t_1, \ldots, t_n \sim$
64 $p_{\boldsymbol{r}}(t)$, the color estimate is gatherd by

$$\hat{C}_{MC}(\boldsymbol{o}, \boldsymbol{d}) = \frac{1}{m}\sum_{i=1}^{m} c(\boldsymbol{o} + t_i \boldsymbol{d}, \boldsymbol{d}). \quad (4)$$

65 Due to the importance sampling with distribution $p_{\boldsymbol{r}}(t)$, each term in Eq 4 contributes equally to the
66 sum as the samples come from regions with non-negligible density. Unlike the grid estimate in Eq. 2,
67 the Monte-Carlo estimate depends on the scene density $\sigma$ implicitly and requires a custom gradient
68 estimate for the parameters of $\sigma$. For instance, NeRF adresses the issue via a hierarchical sampling
69 scheme. It trains a coarse model with a grid approximation to generate importance-weighted ray
70 locations for a separate fine-grained model.

71 In the next section, we propose a propose a novel principled approach to training neural radiance
72 fields with importance-weighted color approximation as in Eq. 4.

## 3 Learning with Stochastic Color Estimates

In this section, we will discuss stochastic approximations to the expected color $C(\boldsymbol{o}, \boldsymbol{d})$ in detail. Recall that $C(\boldsymbol{o}, \boldsymbol{d}) = \mathbb{E}_T c(o + Td, d)$, where $T$ is a random variable with density specified in Eq. 1. Even though density $p_{\boldsymbol{r}}(t)$ involves an integral we cannot compute in closed form, below we first assume that we have an algorithm to compute $\int_{t_n}^{t} \sigma_{\boldsymbol{r}}(s) \mathrm{d}s$ used in $p_{\boldsymbol{r}}(t)$.

Given a groundtruth expected color $C_{gt}$, optimization objective in NeRF captures the difference $L(\hat{C}(\boldsymbol{o}, \boldsymbol{d}), C_{gt})$ between $C_{gt}$ and the estimated color $\hat{C}(\boldsymbol{o}, \boldsymbol{d})$. To reconstruct a scene NeRF runs a gradient based optimizer to minimize the objective averaged across multiple rays and multiple viewpoints. Such approach works for grid estimate $\hat{C}(\boldsymbol{o}, \boldsymbol{d}) = \hat{C}_{Riemann}(\boldsymbol{o}, \boldsymbol{d})$ that depends on density $\sigma_r$ explicitly, but Monte-Carlo estimate $\hat{C}_{MC}(\boldsymbol{o}, \boldsymbol{d})$ of the expectation depends on $\sigma$ implicitly and a naive automatic differentiation algorithm will return zero gradients.

In the rest of the section, we first introduce an algortihm to compute $\hat{C}_{MC}(\boldsymbol{o}, \boldsymbol{d})$ and derive a gradient estimate for the algorithm. Then, we conclude with a discussion our implementation of the estimate. To ease the notation, we will also introduce $\sigma_{\boldsymbol{r}}(t) = \sigma(\boldsymbol{o} + t\boldsymbol{d})$ and $c_{\boldsymbol{r}}(t) = c(\boldsymbol{o} + t\boldsymbol{d}, \boldsymbol{d})$ to denote fields restricted to a ray $\boldsymbol{r} = \boldsymbol{o} + t\boldsymbol{d}$.

### 3.1 Estimate Reparameterization

To make the dependence of $\hat{C}(\boldsymbol{o}, \boldsymbol{d})$ on $\sigma_{\boldsymbol{r}}$ explicit, we change the variables in the expectation $\mathbb{E}_T c_{\boldsymbol{r}}(T)$. For $F(t) = 1 - \exp\left(-\int_{t_n}^{t} \sigma_{\boldsymbol{r}}(s) \mathrm{d}s\right)$ and $y := F(t)$ we write

$$\mathbb{E}_T c_{\boldsymbol{r}}(T) = \int_{t_n}^{+\infty} c_{\boldsymbol{r}}(t) p_{\boldsymbol{r}}(t) \mathrm{d}t = \int_{y_n}^{y_f} c_{\boldsymbol{r}}(F^{-1}(y)) \mathrm{d}y. \tag{5}$$

Function $F(t)$ acts as cumulative distribution function of the variable $T$ with a single exception that, in general, $y_f = \lim_{t \to \infty} F(t) \neq 1$. In volume rendering, $F(t)$ is called the opacity function with $y_f$ being equal to pixel opaqueness. Bounds of integration are where $y_n = F(t_n) = 0$ and $y_f = \lim_{t \to +\infty} F(t)$. For simplicity, below we replace $y_f$ with $F(t_f)$ where $t_f$ is the maximum ray depth.

In the right-hand side of Eq. 5, integration boundaries depend on the opacity $F$ and, thus, on the volume density $\sigma_{\boldsymbol{r}}$. We further simplify the integral by changing the integration boundaries to $[0, 1]$ :

$$\int_{y_n}^{y_f} c_{\boldsymbol{r}}(F^{-1}(y)) \mathrm{d}y = \int_0^1 (y_f - y_n) c_{\boldsymbol{r}}(F^{-1}(y_n + (y_f - y_n)u)) \mathrm{d}u. \tag{6}$$

With this, we arrive to the following reparameterized Monte-Carlo estimate of the expected color obtained with i.i.d $U[0, 1]$ samples $u_1, \ldots, u_m$:

$$\hat{C}_{MC}^R(o, d) := \frac{1}{m} \sum_{i=1}^{m} (y_f - y_n) c_{\boldsymbol{r}}(F^{-1}(y_n + (y_f - y_n)u_i)). \tag{7}$$

In the above estimate sampling does not depend on volume density $\sigma_{\boldsymbol{r}}$ or color $c_{\boldsymbol{r}}$. Essentially, this is a reparameterized Monte-Carlo estimate that generates samples from $p_{\boldsymbol{r}}(t)$ using the inverse cumulative distribution function $F^{-1}(y_n + (y_f - y_n)u)$.

We further improve the estimate using stratified sampling. To do this, we replace the uniform samples $u_1, \ldots, u_m$ with uniform independent samples within regular grid bins $v_i \sim U[\frac{i-1}{m+1}, \frac{i}{m+1}], i = 1, \ldots, m$ and derive a reparameterized (R) stratified (S) Monte Carlo estimate

$$\hat{C}_{SMC}^R(o, d) = \frac{1}{m} \sum_{i=1}^{m} (y_f - y_n) c_{\boldsymbol{r}}(F^{-1}(y_n + (y_f - y_n)v_i)). \tag{8}$$

It is easy to show that both 7 and 8 are unbiased estimates of 1.

Next, we will discuss algorithms used to compute the inverse opacity function $F^{-1}(y)$ and compute the gradients of the function with automatic differentiation.

## 3.2 Implementation of Inverse Opacity for Volume Sampling

To compute the estimates in Eqs. eqs. (7) and (8), we need to compute the inverse opacity $F^{-1}(y)$ along with its gradient. In practice, we start with a black-box density field $\sigma_{\boldsymbol{r}}(x)$ and compute the induced density $p_{\boldsymbol{r}}(t)$ and opacity $F(t)$ on a ray $r$ via approximations. Assuming we have an algorithm to compute $\int_{t_n}^{t} \sigma_{\boldsymbol{r}}(s)\mathrm{d}s$, below we show how to implement the inverse opacity $F^{-1}$.

We invert $F(t) = 1 - \exp\left(-\int_{t_n}^{t} \sigma_{\boldsymbol{r}}(s)\mathrm{d}s\right)$ with binary search. Note that $F(t)$ is a monotonic function and for $y \in (y_n, y_f) = (F(t_n), F(t_f))$ the inverse lies in $(t_n, t_f)$. To compute $F^{-1}(y)$, we start with boundaries $t_l = t_n$ and $t_r = t_f$ and gradually decrease the gap between the boundaries based on the comparison of $F(\frac{t_l + t_r}{2})$ with $y$. Importantly, such procedure is easy to parallelize across multiple inputs and multiple rays.

However, we cannot backpropagate through the binary search iterations and need a workarond to compute the gradient $\frac{\partial t}{\partial \theta}$ of $t(\theta) = F^{-1}(y, \theta)$. To do this, we compute differentials of the right and the left hand side of equation $y(\theta) = F(t, \theta)$

$$\frac{\partial y}{\partial \theta}\mathrm{d}\theta = \frac{\partial F}{\partial t}\frac{\partial t}{\partial \theta}\mathrm{d}\theta + \frac{\partial F}{\partial \theta}\mathrm{d}\theta. \tag{9}$$

By the definition of $F(t, \theta)$ we have

$$\frac{\partial F}{\partial t} = (1 - F(t, \theta))\sigma_{\boldsymbol{r}}(t, \theta), \tag{10}$$

$$\frac{\partial F}{\partial \theta} = (1 - F(t, \theta))\frac{\partial}{\partial \theta}\left(\int_{t_n}^{t}\sigma_r(s, \theta)\mathrm{d}s\right). \tag{11}$$

We solve Eq. 9 for $\frac{\partial t}{\partial \theta}$ and substitute the partial derivatives using Eqs. eqs. (10) and (11) to obtain the final expression for the gradient

$$\frac{\partial t}{\partial \theta} = \frac{\frac{\partial y}{\partial \theta} - (1 - F(t, \theta))\frac{\partial}{\partial \theta}\int_{t_n}^{t}\sigma_{\boldsymbol{r}}(s, \theta)\mathrm{d}s}{(1 - F(t, \theta))\sigma_{\boldsymbol{r}}(t, \theta)}. \tag{12}$$

In our implementation, we use automatic differentiation to compute $\partial y / \partial \theta$ and $\frac{\partial}{\partial \theta}\int_{t_n}^{t}\sigma(s)\mathrm{d}s$ to combine the results as in Eq. 12.

## 3.3 Computing Opacity in Practice

To describe the sampling procedure, we assumed that we have an oracle for computing $\int_{t_n}^{t}\sigma_r(s)\mathrm{d}s$ along with its gradient. The integral is required to compute opacity $F(t)$. In this work, we consider an arbitrary volumetric density $\sigma(s)$ and approximate it with a linear spline on a ray $r = o + td$ to sample the points on the ray. Specifically, we take a grid $t_0 < \cdots < t_m$ and compute $\sigma_r(t_0), \ldots, \sigma_r(t_n)$ to construct the spline $\hat{\sigma}_r(s)$ (Fig. 1). For the piecewise linear function $\hat{\sigma}_r(x)$ we can compute the integral $\int_{t_n}^{t}\hat{\sigma}_r(s)\mathrm{d}s$ in a closed form. Additionally, we can backpropagate the gradients through the approximation to compute the gradients of knots $\sigma_r(t_0), \ldots, \sigma_r(t_m)$. Thus, we obtain a differentiable rendering algorithm for an arbitrary density field $\sigma$. Besides that, some recent works parameterize

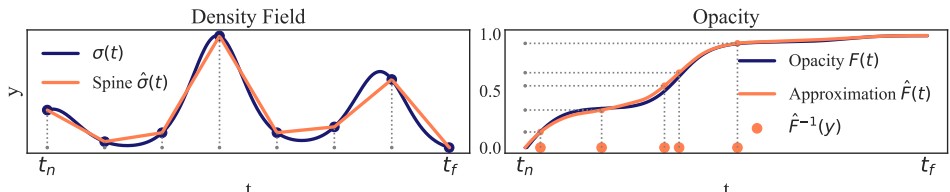

Figure 1: Illustration of opacity inversion. We approximate an arbitrary density field $\sigma$ with a linear spline(left). Then we use the spline to approximate opacity $\hat{F}(t)$ and compute $\hat{F}^{-1}(y)$ (right).

density fields thruogh voxel grids. For a voxel grid, when $\sigma_{\boldsymbol{r}}$ is a trilinear interpolation of the grid values, we can compute the integral in a closed form.

## 4 Related Work

**Neural Radiance Fields & Efficient Sampling** Even in the orginial work on neural radiance fields [8] the authors aimed to find an efficient sampling algorithm for volume rendering. Our importance sampling approach is reminscent of their hierarchical sampling solution. On the first stage, they use an auxilliary model on a sparse grid. Then they use the predicted densities to generate a dense grid with a improtance sampling-like algorithm. As opposed to NeRF, we compute density on a dense grid at the first stage and then use a sparse set of samples to evaluate radiance on the second stage. Our algorithm also allows training without auxilliary models.

Several recent follow-up works also aimed to improve NeRF rendering time and overall efficiency. Most of these works consider trainable encoding $\theta$ and utilize some efficient data structure to make each evaluation of multi-layered perceptron fast or avoid evaluating MLP at all. One of the earliest work in this direction was NSVF [7]. The authors proposed to use octree to store point-based embeddings and then estimate query point embedding with a trilinear interpolation and positional encoding. During training, the octree gradually increased resolutionn in the regions of interest and pruned the empty areas. However, this method still requires the time-consuming training of MLPs. Voxel-based embedding structure was further studied in recent works and it was shown that positional encoding doesn't affect model convergence - the network can be trained with fully trainable embedding without any encoding. And also, what is more important, such a structure allows for making neural network (MLP) shallower and consequently faster. Following this idea, DirectVoxGo [12] proposes to avoid MLP at all in density computation, while Instant Neural Graphic Primitives [10] uses it to solve hash collisions. When density field is a piecewise linear we can compute opacity in a closed-form.

**Reparameterization Trick & Implicit Differentiation** Our solution is inspired by the literature on deep latent variable models [6, 11] and approximate inference. In this area, models often contain an internal sampling algorithm with parameters we need to optimize. The now-common approach for continuous random variables is the reparameterization trick, which we apply in our setup. The authors of [9] give a comprehensive overview of the area state.

A closely related work in the context of deep variable models is [4]. They were first to apply implicit differentiation to estimate gradients for the reparameterization trick. While we use the implicit differentiation to compute the gradient of binary search output, the same approach applies to other iterative algorithms. The examples include ODE solves [3], fixed-point iterators [1] and optimization algorithms.

## 5 Experiments

### 5.1 Importance Sampling for a Single Ray

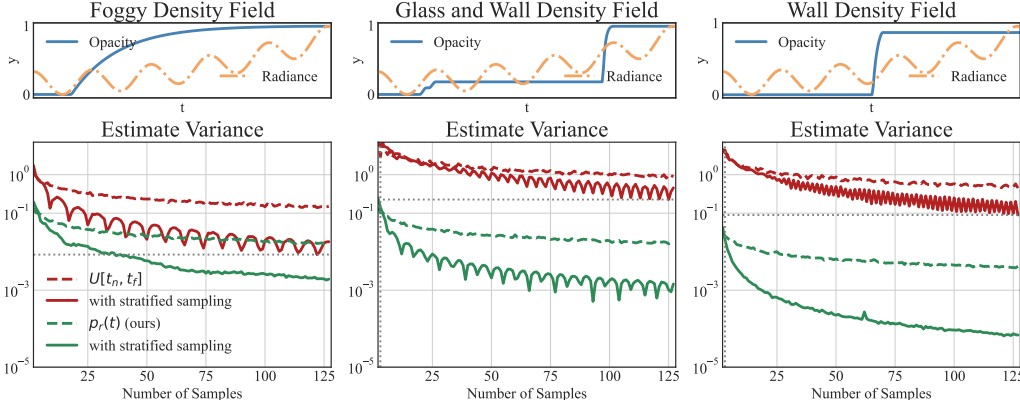

Figure 2: Color estimate variance compared for a varying number of samples. The upper plot illustrates underlying opacity function on a ray; the lower graph depicts variance in logarithmic scale. Our importance sampling approach (solid green) has significantly lower deviation than a stratified baseline (solid red) typically used in volume rendering.

We begin with an evaluation of color estimates in a one-dimensional setting. Our experiment models light propagation on a single ray in three typical situations. The upper row of Fig. 2 defines a scalar radiance field (orange) $c(t)$ and opacity functions (blue) $F(t)$ for

- "Foggy" density field. It models a semi-transparent volume. Similar fields occur after model initialization during density field training;

- "Glass and wall" density field. Models light passing through nearly transparent volumes such as glass. The light is emitted at three points: the inner and outer surface of the transparent volume and an opaque volume near the end of the ray;

- "Wall" density field. Light is emitted from a single point on a ray. Such fields are most common in applications.

For the three fields we estimated the expected radiance $C = \int_{t_n}^{t_f} c(t) \mathrm{d}F(t)$. We considered two baseline methods (both in red in Fig. 2): the first was a Monte Carlo estimate of $C$ obtained with $U[t_n, t_f]$ samples, the second was a stochastic modification of Eq. 2 using a grid $t_n = t_0 < \cdots < t_m = t_f$:

$$\hat{C} = \sum_{i=1}^{m} (t_i - t_{i-1}) c(\tau_i) \frac{\mathrm{d}F}{\mathrm{d}t}\bigg|_{t=\tau_i}, \text{ with independent } \tau_i \sim U[t_i - 1, t_i]. \tag{13}$$

In other terms, the second baseline uses stratified sampling to reduce the baseline Monte Carlo estimate variance. Eq 13 is an instance of a vanilla volume rendering algorithm one may encounter in practice. We compared the baseline against estimate from Eq. eq. (7) and its stratified counterpart from Eq. 8. All estimates are unbiased. Therefore, we only compared the estimates variances for a varying number of samples $m$.

In all setups, our stratified estimate uniformly outperformed the baselines. For the most challenging "foggy" field, approximately $m = 32$ samples we required to match the baseline performance for $m = 128$. We matched the baseline with only a $m = 4$ samples for other fields. Importance sampling requires only a few points for degenerate distributions. In further experiments, we take $m = 8, 32$ to obtain a precise color estimate even when a model did not converge to a degenerate distribution.

## 5.2 Scene Reconstruction with Reparameterized Volume Sampling

Next, we apply our algorithm to 3D scene reconstruction based on a set of image projections. As a benchmark, we use the common Lego dataset. The primary goal of the experiment is to demonstrate computational advantages of our algorithm compared to a basic volume rendering baseline.

As a starting point, we took the original NeRF's MLP [8] with eight layers used to compute density and radiance. Then we modified the architecture to use only three first layers to compute the density field. When the density field is queried, we only compute the first three layers, while for the radiance we compute the whole network. Even though such modification may have put additional limitations on the density model, it illustrates the benefit of using fewer radiance queries. For density, we used softplus activation to ensure its positivity, while for the radiance we used sigmoid activation to ensure that the output will be a valid RGB image.

In our experiment, we did not reproduce the expensive hierarchical sampling used in NeRF and trained a single model in all experiments. Our baseline calculated color using Eq. **??**. We took a dense grid of $m = 128$ points along each ray and trained the model using Huber loss with the ground truth colors and the predict colors. We additionally perturbed the grid to r egularize the model. We used Adam [5] optimizer for training and decayed the learning rate during 100 epochs of training from $3\mathrm{e}{-4}$ to $3\mathrm{e}{-7}$ following MIP-NeRF's scheduler [2] with image batch size equal 8 and each epoch consisting of 8000 batches. To form a training batch, for each image in an image batch we selected 375 pixels and cal culated loss over them.

We evaluated the importance sampling-based rendering algorithm with the same architecture and hyperparameters as with the baseline model. We used the same algorithm to sample a dense grid of $m = 128$ points to query the density field and construct an approximating spline. Then we calculated color approximation with Eq. 8 with $m' = \{8, 32\}$ samples from the inverse cumulative density function approximated by the spline.

| Model | PSNR ($\uparrow$) | SSIM ($\uparrow$) | LPIPS (alex) [14]($\downarrow$) |
|---|---|---|---|
| Baseline | 27.247 | 0.904 | 0.1138 |
| Splines, #pts in estimation 8 | | | |
| Training      Validation | | | |
| 8 pts      1 pts | 23.377 | 0.822 | 0.1819 |
| 8 pts      2 pts | 25.193 | 0.858 | 0.1449 |
| 8 pts      4 pts | 26.210 | 0.883 | 0.1215 |
| 8 pts      8 pts | 26.502 | 0.892 | 0.1243 |
| 8 pts      16 pts | 26.570 | 0.894 | 0.1333 |
| 8 pts      32 pts | 26.585 | 0.895 | 0.1369 |
| 32 pts      1 pts | 22.519 | 0.805 | 0.2050 |
| 32 pts      2 pts | 24.902 | 0.846 | 0.1523 |
| 32 pts      4 pts | 26.523 | 0.881 | 0.1181 |
| 32 pts      8 pts | 27.100 | 0.897 | 0.1083 |
| 32 pts      16 pts | 27.252 | 0.902 | 0.1167 |
| 32 pts      32 pts | 27.286 | 0.904 | 0.1223 |

Table 1: Ablation study and comparison with the baseline. Metrics are calculated over test views for Lego scene [8]

First, we compared the rendering quality of our algorithm against the baseline. Tab. 1 contains the quantitative results and figs. 3 and 4 contain qualitative results. From the rendering quality viewpoint (1), with $m' = 32$ samples, our model works on par with the baseline, while with $m' = 8$ samples it has slightly worse performance. Though we did not aim to reproduce the state-of-the-art results, we speculate that a better density model could improve the results even further. In Fig. **??**, we compared the rend ering performance of importance sampling for varying $m'$. Our algorithm produced sensible renders even for $m' = 1$, however noise artifacts only disappeared for $m' = 32$. Fig. 4 shows a stratified estimate renders (Eq. 8) along with a Monte Carlo renders (Eq. 7) for $m' = 32$. With the same rendering complexity, the variance reduction obtained via stratified sampling purges the rendering artifacts that a naive Monte Carlo estimate has.

| Model | Iter/sec ($\uparrow$) | Mem Usage ($\downarrow$) |
|---|---|---|
| Baseline | 3.90 | 8.5 Gb |
| Splines 1 pts | 4.89 | 1.8 Gb |
| Splines 2 pts | 5.06 | 1.8 Gb |
| Splines 4 pts | 4.88 | 2.1 Gb |
| Splines 8 pts | 4.53 | 2.2 Gb |
| Splines 16 pts | 3.81 | 2.5 Gb |
| Splines 32 pts | 2.98 | 2.8 Gb |

Table 2: Speed & memory estimates. Iteration time is measured during training on GTX 1080 ti, memory usage is measured during inference with batch size equal 1024

Besides the rendering quality, we estimated the training speed and memory footprint of our algorithm in Tab. 2. For $m' = 8$ training iterations were on average $\times 1.2$ faster, while for $m' = 32$ training iterations took $\times 1.3$ more time. The difference occurred due to a varying number of radiance queries. For a memory footprint viewpoint, our algorithm used $\times 3.0$ and $\times 3.9$ less memory for $m' = 32$ and $m = 8$ correspondingly. With this, important sampling leaves room for further optimization as it allows to work with bigger batches with a moderate variability in rendering speed and quality.

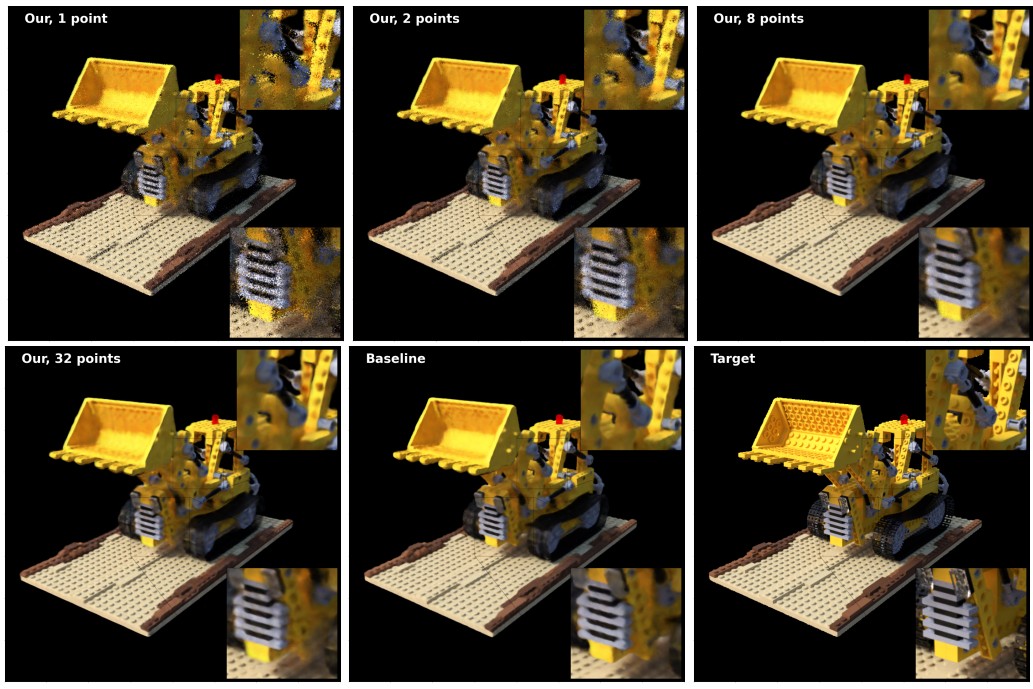

Figure 3: Rendering results with a different number of samples in the stratified estimate. From left to right and from top to down: 1, 2, 8, 32 points estimates, Baseline and Target for reference.

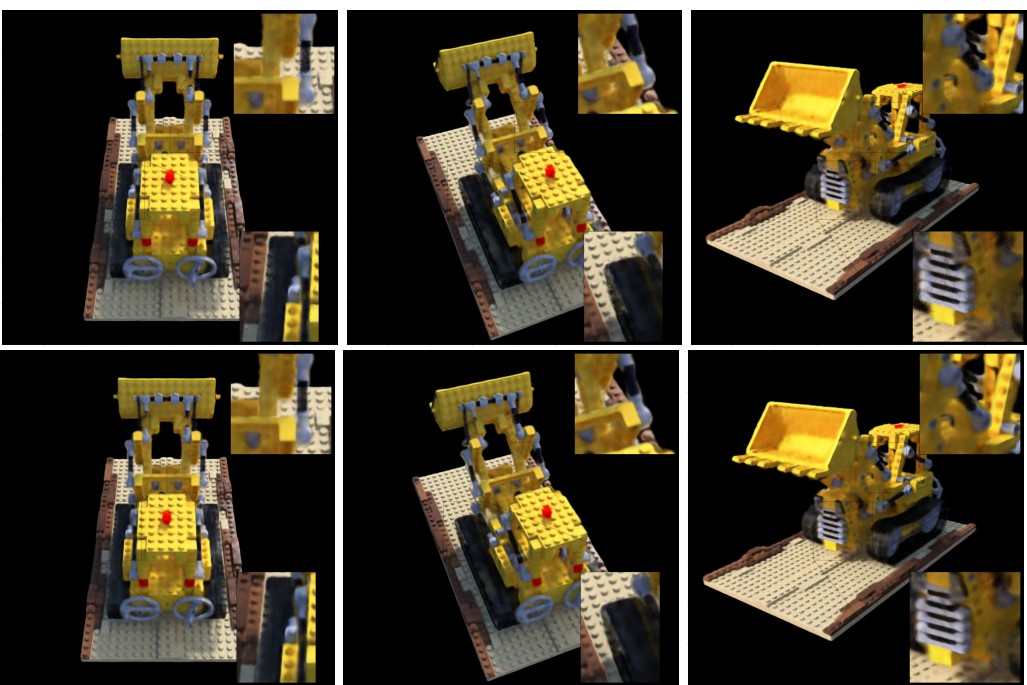

Figure 4: Comparison of rendering results from different viewing angles with Monte-Carlo estimate (top row) and stratified Monte-Carlo estimate (bottom row), both with 32 points along each ray

# 6   Conclusion

We proposed an alternative to classic volume rendering algorithms used in 3D scene reconstruction. For a synthetic experiment and in full-scale reconstruction task we achieve better estimation results in terms of variance with a significantly smaller computation footprint. In particular, our algorithm allows for significant memory reductions and even increased inference time. At the same time, we demonstrate competitive rendering quality. We believe that our approach is a promising altenative to standard volume rendering techniques.

## 6.1   Broader Impact

We hypothesize that models like NeRFs may be used in online stores for a better user experience. Then people will choose more suitable products. We are not aware of any possibilities to use this in a negative way. Furthermore, we are sure that the efficient sampling we proposed for 3D rendering may reduce computation costs and therefore environmental damage.

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
