# OpenReview forum: "Differentiable Rendering with Reparameterized Volume Sampling"
_NeurIPS.cc/2022/Conference — NeurIPS 2022 Submitted_

### Official Review · Reviewer_2V6p · 2022-07-10

**Rating:** 4
**Confidence:** 3
**Soundness:** 3 good
**Presentation:** 3 good
**Contribution:** 2 fair

**Summary:**

A Monte Carlo sampling strategy is proposed for rendering NeRF-based density field representation. The method works by firstly densely querying the density field to compute an inverse CDF, which is later used to reduce the number of subsequent samples of colour network.

The method is compared to the NeRF baseline and achieved faster speeds (~20%) but at the cost of slightly worse image quality (~0.7dB reduction in PSNR).

**Questions:**

I would suggest the authors to include missing references, and expand the experiment section.

To me the main drawback of proposed method is that dense sampling of density network still takes significant time. As such the speed gain is rather limited compared to vanilla NeRF. I wonder whether using a smaller density network can help in this regard.

**Limitations:**

I would suggest discussing the limitations in detail in section 6.

**Strengths And Weaknesses:**

The paper does address an important problem, that is to improve the efficiency of NeRF rendering. The proposed method is well motivated and indeed effective in that it does significantly reduce the sampling rate of colour network (8 to 32 samples per ray compared to 128 samples per ray in NeRF). However, on the other hand, it still requires a dense sampling of density network in the first stage, and as such the overall rendering efficiency is only slightly, if at all, improved without sacrificing performance.

Many related papers are missing from related work section and should be cited and compared to, these include faster NeRFs:
- David B Lindell, Julien NP Martel, and Gordon Wetzstein. 2021. Autoint: Automatic integration for fast neural volume rendering
- Daniel Rebain, Wei Jiang, Soroosh Yazdani, Ke Li, Kwang Moo Yi, and Andrea Tagliasacchi. 2021. Derf: Decomposed radiance fields.
- Stephan J. Garbin, Marek Kowalski, Matthew Johnson, Jamie Shotton, and Julien Valentin. 2021. FastNeRF: High-Fidelity Neural Rendering at 200FPS
- Alex Yu, Ruilong Li, Matthew Tancik, Hao Li, Ren Ng, and Angjoo Kanazawa. 2021. Plenoctrees for real-time rendering of neural radiance fields.
- Christian Reiser, Songyou Peng, Yiyi Liao, and Andreas Geiger. 2021. KiloNeRF: Speeding up Neural Radiance Fields with Thousands of Tiny MLPs.

There are several papers on implicit surface volumetric sampling, that may be worth citing as well:
- Peng Wang, Lingjie Liu, Yuan Liu, Christian Theobalt, Taku Komura, and Wenping Wang. 2021. NeuS: Learning Neural Implicit Surfaces by Volume Rendering for Multi-view Reconstruction.
- Michael Oechsle, Songyou Peng, Andreas Geiger. 2021. UNISURF: Unifying Neural Implicit Surfaces and Radiance Fields for Multi-View Reconstruction.

Experiments are too limited. There is only one case being the lego scene. Table 1 only includes two training sampling rates, what are the results for 1,2,8,16 samples per ray?

Some typos need to be corrected, e.g.:
line 71: propose a propose a
line 208, 224: broken ref

---

### Official Review · Reviewer_46SP · 2022-07-11

**Rating:** 3
**Confidence:** 4
**Soundness:** 1 poor
**Presentation:** 1 poor
**Contribution:** 1 poor

**Summary:**

The paper proposes a reparametrization of volume sampling for faster rendering of neural radiance fields. The key idea is that, when the density value (i.e., \sigma) is known at a few sparse points, volume rendering can be reparametrized for importance sampling, which reduces the variance and thereby accelerates the rendering of the neural radiance field. The paper provides the demonstration of lower variance (compared to uniform sampling), ablation study, and novel-view synthesis with a different number of samples both qualitatively and quantitatively.

**Questions:**

Why is the proposed method efficient compared to the hierarchical volume sampling of the original NeRF paper? There is no clear demonstration of this. Both require the coarse sampling stage, which is the same additional cost compared to the naive sampling strategy. If the proposed method modifies the network structure to reduce this additional computational burden, the original NeRF can also benefit from this modification. The main baseline should be the original NeRF paper, and the comparison should be stated much more clearly and fairly in the experiment.

**Limitations:**

The checklist says the limitations are described in Section 1, but Section 1 does not include any description of the limitation.

**Strengths And Weaknesses:**

While this paper tackles an interesting and important problem in neural rendering, which is an acceleration of NeRF rendering, it needs to be much improved in terms of presentation and evaluation.

- The proposed method is not evaluated properly. The key contribution of this paper is importance sampling that can accelerate NeRF rendering, but the provided evaluation does not demonstrate this contribution. Figure 2 shows the variance compared to uniform sampling, but the baseline should be the importance sampling used in the original NeRF. Any importance sampling should provide better results than uniform sampling, and this experiment cannot show any meaningful demonstration as the proposed method has an additional cost of density sampling in the previous step for the reparametrization. Also, in Section 5.2, the proposed method modified the original NeRF to reduce the computational cost of density sampling (by changing the density network from eight layers to three layers), which will reduce the quality of the density field and make the resultant comparison NOT apple-to-apple. Moreover, the evaluations are conducted only on a single scene (i.e., lego) and the improvement over the baseline is marginal even in the current evaluation.
- The presentation can be much improved. There are a number of typos and the notation is not consistent throughout the paper, which severely reduces the clarity of the paper considering the fact that the main contribution is in the reparametrization of equations. To list a few:
    - Line 47: R^3 → \mathbb{R}^3
    - Line 48: directionn → direction
    - Line 48: location x → \mathbf{x}, direction d → \mathbf{d}
    - Line 53: ray r → ray \mathbf{r}
    - Line 58: fauithfull → faithful
    - Line 75: c(o + Td, d) → c and d should be bold
    - Line 82: \sigma_r → both \sigma and r should be bold
    - Notation for Equations are not consistent; there are Eq n, Eq. n, n, Eq. eq. (n).
    - Lines 110, 123: Eqs. eqs
    - Line 130: r = o + td → r, o, d, should be bold
    - Line 208: Eq. ??
    - Line 224: Fig. ??
    - Line 225: ren dering → rendering
- The paper is based on Equation (2), which is the Riemann sum for the integral of volume rendering. However, it is not directly equivalent to the discrete volume rendering used in the original NeRF paper (Equation (3) in the original NeRF paper that is based on [Max 1995]) because of the small difference in the derivation (e.g., density is locally uniform between the samples). The difference in the equation should be clarified for a better comparison to the original NeRF.

---

### Official Review · Reviewer_jLZr · 2022-07-11

**Rating:** 3
**Confidence:** 4
**Soundness:** 3 good
**Presentation:** 1 poor
**Contribution:** 2 fair

**Summary:**

This submission proposes a differentiable importance sampling to estimate the rendering equation used in NeRF based methods. The method is derived using the re-parameterization method. The gradient descent is derived based on the re-parameterization method. Compared with the previous sampling method used in NeRF based methods, the proposed method leads to much lower sampling variance under the same computational cost.


**Questions:**

For spline construction (L128-L130), what if the underlying density field is not piecewise linear, or have sharp spikes along a ray (eg. thin structure)? Will the spline approximation still be accurate enough to get opacity estimation?

L131-133: how to get the sampled grid t_0<...<t_m in the first place? Are they predefined? If so, how to guarantee that they don’t miss any non-empty space across different views and across different scenes?

**Limitations:**

Experiment is performed on only one scene. So it’s unknown whether this method is finetuned for that scene or more general.

From L131-133, the sampling grid t_0<...<t_m cannot be guaranteed to cover the non-empty space.

**Strengths And Weaknesses:**

Strength

+The proposed importance sampling for Monte-Carlo (MC) estimate is technically sound by being based on the re-parameterization method.

+The author also shows how to make the importance sampling along the ray differentiable by taking advantage of the re-parameterization. As a result, the new sampling method can be plugged into the neural rendering pipeline and make the implicit function trainable end-to-end.

+The proposed differentiable sampling method has much lower sampling variance as compared to the existing sampling method used in the NeRF-based methods.

+The density field is approximated by splines in closed form. This makes the integration along the ray for getting the light attenuation more efficient to compute, as compared to the previous methods where another integration is performed along the ray.

Weakness

-Although the proposed sampling method and its corresponding backpropagation process is sound and promising, the paper lacks the experiments to validate its effectiveness:

1.The method is only evaluated on a single scene (Lego)

2.The method is only compared to the base NeRF. There are works on speeding up NeRF without sacrificing performance during both training and rendering (eg. Plenoxel NeRF). How does this method compare those methods? And more importantly, can the method be plugged into those methods for further speed-up? Without the corresponding experiment, those questions can hardly be answered.

3.Key experiment missing for trade-off between speed and reconstruction quality: How is reconstruction quality degrades with the decrease # of samples (ie. speed-up) by using the proposed method?

4.Related to the missing speed-quality trade-off: From Tab.2, the estimation time per iteration  increases with the number of points in the splines and exceeds the baseline at around 16 points. So the question is how does the reconstruction quality compare with the baseline at around 16 sampling points. The paper does not include the quality comparison .


5.Comparison to other SOTA method missing: How is the proposed method compared to the SOTA NeRF methods in terms of reconstruction quality and computational and memory consumption


-Paper needs more refinement: There are multiple question marks in the paper due to error in reference.

---

### Author Response · Authors · 2022-08-02
**Official Comment (1/3)**

We thank reviewers for their time and thought-provoking questions. We have improved the method evaluation according to the reviews. Below we will present the updated results and address some of the major concerns raised by the reviewers.

# Experiments
## Evaluation of a single scene (jLZr, 46SP, 2V6p)

In the initial submission, we focused on the memory performance but did not demonstrate whether the method reproduces the results obtained with the standard integration methods. To eliminate this shortcoming, we ran the experiments with an architecture identical to the original NeRF architecture on all synthetic scenes.

As in the non-hierarchical model, we used 256 density evaluations to construct the spline. We used 32 samples to estimate the expected color given the spline during training. As we do not run the color layers at the density evaluation stage and do not run the density layers while evaluating the radiance, such a configuration has a memory footprint comparable to a non-hierarchical baseline. At inference, we used 64 radiance samples. We used Softplus activation with $\beta = 10$ for the density instead of ReLU used in NeRF. Other hyperparameters were identical to the ones used in NeRF.

In the table below, we report the obtained results. For reference, we also provide the NeRF metrics with and without hierarchical sampling.

| PSNR↑                | Chair  | Drums  | Ficus  | Hotdog | Lego  | Materials | Mic   | Ship  | Avg   |
|----------------------|--------|--------|--------|--------|-------|-----------|-------|-------|-------|
| NeRF full            |  33.00 |  25.01 |  30.13 |  36.18 | 32.54 |   29.62   | 32.91 | 28.65 | 31.01 |
| NeRF no hierarchical |  31.32 |  24.55 |  29.25 |  35.24 | 31.42 |   29.22   | 31.74 | 27.73 | 30.06 |
| Ours                 |  31.35 |  22.42 |  28.42 |  34.36 | 30.70 |   28.72   | 31.18 | 26.89 | 29.26 |

| SSIM↑                | Chair  | Drums  | Ficus  | Hotdog | Lego  | Materials | Mic   | Ship  | Avg   |
|----------------------|--------|--------|--------|--------|-------|-----------|-------|-------|-------|
| NeRF full            |  0.967 |  0.925 |  0.964 |  0.974 | 0.961 |   0.949   | 0.980 | 0.856 | 0.947 |
| NeRF no hierarchical |  0.951 |  0.914 |  0.956 |  0.969 | 0.951 |   0.944   | 0.973 | 0.844 | 0.938 |
| Ours                 |  0.956 |  0.875 |  0.949 |  0.965 | 0.946 |   0.940   | 0.971 | 0.824 | 0.928 |

| LPIPS↓ (VGG)         | Chair  | Drums  | Ficus  | Hotdog | Lego  | Materials | Mic   | Ship  | Avg   |
|----------------------|--------|--------|--------|--------|-------|-----------|-------|-------|-------|
| NeRF full            |  0.046 |  0.091 |  0.044 |  0.121 | 0.050 |   0.063   | 0.028 | 0.206 | 0.081 |
| NeRF no hierarchical |  0.065 |  0.177 |  0.056 |  0.130 | 0.072 |   0.080   | 0.039 | 0.249 | 0.109 |
| Ours                 |  0.065 |  0.178 |  0.066 |  0.078 | 0.083 |   0.077   | 0.040 | 0.225 | 0.102 |

Our method is worse than NeRF without hierarchical sampling in terms of average PSNR and SSIM, although it is slightly better in terms of average LPIPS. As we have only modified the underlying integration scheme, we expected the model performance to match the non-hierarchical NeRF. After all, both models have the same level of grid granularity. But the experiment did not meet our expectations. As reviewer 46SP pointed out, we use a slightly different integral for expected color evaluation: NeRF aggregates the samples with weights given by a discrete distribution whereas we use a continuous aggregation model. We speculate that this may be the reason for the performance gap and can be addressed with a better selection of the density activation layer. Still, both models are indistinguishable with a bare eye.

At the same time, our method is worse than hierarchical NeRF. We argue that comparison against the two-stage NeRF in terms of quality would not be entirely fair. Specifically, the hierarchical model uses an auxiliary network for importance sampling and aggregates the produced samples with weights given by a second "fine" model. These two models operate on different scales and help to improve the effective model resolution (i.e. coarse model divides a ray into $m_{coarse} = 64$ parts and then fine model may split a single bin into $m_{fine} = 128$ parts). Compared to the full NeRF model, we only use a single network for importance sampling and do not re-weight the produced samples. Therefore, the underlying spline resolution of $m = 256$ points per ray limits the effective resolution of the 3D model. The lack of sample re-weighting is an apparent limitation of the experiment. We plan to address it in future work.

---

> ### Author Response · Authors · 2022-08-02
> **Official Comment (2/3)**
>
> ## Render quality w.r.t. number of samples at inference (jLZr, 2V6p)
>
> We evaluated the proposed method with a varying number of samples at the inference stage. Specifically, we took the NeRF model from the above experiment and computed PSNR, SSIM, and LPIPS on the Lego scene test set. Additionally, we report the rendering time of a single image with a batch consisting of 4k rays. The underlying model used 32 samples to estimate the expected ray color during the training stage. We observed that all metrics gradually improve with the number of points and stabilize at approximately 16 points.
>
> |               | 1 pt  | 2 pts | 4 pts | 8 pts | 16 pts | 32 pts |  64 pts | NeRF full | NeRF no hierarchical |
> |---------------------|-------|-------|-------|-------|--------|--------|---------|-----------|----------------------|
> | PSNR↑               | 25.67 | 28.30 | 29.96 | 30.54 |  30.70 |  30.74 |   30.70 |    32.54  |         32.42        |
> | SSIM↑               | 0.870 | 0.909 | 0.934 | 0.943 |  0.945 |  0.946 |   0.946 |    0.949  |         0.944        |
> | LPIPS↓              | 0.177 | 0.151 | 0.119 | 0.093 |  0.084 |  0.080 |   0.083 |    0.064  |         0.080        |
> | Rendering time↓ (s) | 20.73 | 20.77 | 20.97 | 21.39 |  22.28 |  24.09 |  27.726 |   36.451  |        24.067        |
> | Memory↓ (mb)        |  5388 |  5388 |  5390 |  5390 |   5394 |   5394 |    5406 |     7757  |          6742        |
>
>
> ## Computational Advantages of Reparameterized Sampling (jLZr, 46SP, 2V6p)
>
> Our sampling method requires far fewer radiance field calls compared to the quadrate rules used in modern NeRF variations. In the initial submission, we used a shallow network for the density field and a deep network for the radiance field to highlight this feature. We obtained a significant memory reduction compared to standard algorithms (Table 2) due to fewer radiance calls. However, such experiment design led to subpar performance compared to SoTA view synthesis methods.
>
> To make a more convincing illustration, we replaced the modified NeRF architecture with a recently proposed Direct Voxel Go (DVGO) solution. DVGO parameterizes the density field with a voxel grid and models the radiance field with a combination of a voxel grid and a neural network. As a result, density field call is almost instant, whereas the radiance field call takes time and memory.
>
> DVGO does not employ hierarchical sampling as in NeRF. Instead, the authors divide training into two stages to improve the overall training time. In the first *(coarse)* stage, the authors train a coarse density grid. On the second *(fine)* stage, the algorithm skips empty ray regions and optimizes a finer grid. The second stage relies on the coarse grid from the first stage to determine which of the ray regions are empty.
>
> We took the official DVGO repository and replaced the quadrature rule with the expected color estimate proposed in our work. Notably, the adaptation only required replacing the ```forward``` method of the model. Below we report the average number of radiance calls for both stages and standard metrics averaged across all Blender synthetic scenes.
>
> |                        | PSNR↑ | SSIM↑ | LPIPS↓ |
> |------------------------|-------|-------|--------|
> | DVGO (Pytorch + CUDA)  | 31.95 | 0.957 |  0.053 |
> | Ours (32 pts, Pytorch) | 30.13 | 0.944 |  0.070 |
> | Ours (8 pts, Pytorch)  | 30.06 | 0.941 |  0.075 |
>
> |              | Radiance @ Coarse ↓ | Radiance Calls @ Fine ↓ | Memory @ Coarse ↓ | Memory @ Fine ↓ | t @ Coarse ↑ | t @ Fine ↑ |
> |------------------------|---------------------|-------------------------|-------------------|-----------------|--------------|------------|
> | DVGO (Pytorch + CUDA)  |         130.7 calls |              13.4 calls |           4702 mb |         5072 mb |   119.0 it/s | 112.4 it/s |
> | Ours (32 pts, Pytorch) |          32.0 calls |              32.0 calls |           4736 mb |         7484 mb |    63.3 it/s |  20.7 it/s |
> | Ours (8 pts, Pytorch)  |           8.0 calls |               8.0 calls |           4718 mb |         5722 mb |   106.4 it/s |  31.2 it/s |
>
> Our sampling algorithm allows us to reduce the number of radiance calls on both coarse and fine stages. We trained our model with 32 and 8 radiance samples. Currently, the reduction comes at the price of slightly worse model quality. We also provide the number of training iterations per second and memory consumptions for reference. Importantly, these numbers **are not** directly comparable. In particular, DVGO employs custom CUDA kernels to improve rendering speed and optimizes rendering for sparsity on the "fine" training stage. Our current implementation does not optimize for sparsity in the DVGO training setup; as a result, we only obtain competitive speed at the non-sparse "coarse" stage. Still, these optimizations apply to our algorithm as well. We plan to adapt them in the future revision to translate the radiance call improvements into the wall time and memory footprint.

---

> > ### Author Response · Authors · 2022-08-02
> > **Official Comment (3/3)**
> >
> > # Method Motivation and Details
> > ### Why is the proposed method efficient compared to the hierarchical volume sampling of the original NeRF paper? If the proposed method modifies the network structure to reduce the computational burden, can the original NeRF also benefit from this modification? (46SP)
> >
> > Indeed, our rendering algorithm is two-stage like the hierarchical rendering algorithm in NeRF. Also, during the inference, one could first evaluate the density field and then compute the radiance only for the ray positions within the top-k density values. It will lead to the same computational benefit at inference. However, such a filtering procedure is not differentiable and does not apply during training as the gradient will not propagate to the positions outside of top-k. Note that the original full NeRF evaluates radiance at each ray point at the training stage. Instead, we propose a principled alternative algorithm that allows determining the number of radiance evaluations per ray in advance. Besides, full NeRF uses an auxiliary loss to tune the coarse model for the hierarchical sampling. The auxiliary loss is necessary because the importance sampling used in the full NeRF model is not differentiable. In contrast, our algorithm is differentiable and allows training via back-propagation.
> >
> > ### The main baseline should be the original NeRF paper, and the comparison should be stated much more clearly and fairly in the experiment. (46SP)
> >
> > We added the original NeRF as a baseline in the experiment described above. Even though our rendering procedure is two-stage as is the rendering procedure in the full NeRF model, the full NeRF model consists of two models of different resolutions whereas our method leads to a single model. Therefore, we argued above that a fair comparison would be between our method and NeRF without hierarchical sampling.
> >
> > ### Will the spline approximation still be accurate enough if the underlying density field is not piecewise linear? How to get the sampled grid $t_0 < ... < t_m$? (jLZr)
> >
> > In our experiments, we sampled spline knows (almost) uniformly across the ray. Specifically, we used a uniform grid with additional noise to regularize the training procedure:
> >
> > ```t = torch.linspace(near, far, m) + (far - near) * torch.rand(m) / (m - 1)```
> >
> >  In this case, the approximation error for $\sigma(t) \in C^2$ (the difference between spline $\hat\sigma(t)$ and the underlying $\sigma(t)$ field at $t$) can be bounded by $\tfrac{C}{m^2}$ via standard calculus arguments. The constant $C$ is proportional to $\max_{t} \sigma''(t)$.
> >
> > From a theoretical perspective, for sufficiently large $m$ spline approximation error will be negligible. The Monte Carlo estimation error for the expected color also gradually decreases with the number of samples (as it does in Figure 2). In practice, we observed deterioration in models with fine details (such as Ship and Drums synthetic models) for $m = 256$. Indeed, the model seems to miss scene details between the spline knots. To address the issue it is possible to take higher $m$ in our method or employ a more sophisticated sampling procedure (as in the full NeRF model).
> >
> > ### Missing references, limited experimental section (2V6p)
> >
> > We are grateful for the list of missing references. Sadly, the current draft was a rushed paper. We will make sure to expand the related work and discussion sections to incorporate these works.

---

> > > ### Comment · Reviewer_46SP · 2022-08-10
> > > **Response to authors**
> > >
> > > Thank you for the response with extensive additional experiments. However, the concerns addressed by reviewers are not clearly addressed in the rebuttal, especially about the difference in the integral for color evaluation and computational advantage. The broken references and typos are also not corrected, which reduces the clarity of the paper.
> > >
> > > As the authors pointed out, the difference between the integral in NeRF and that in this paper should be further explored and clarified. In the current form, the paper is “approximating the integral” in a different way, not “reparametrizing” the original equation, which is totally different from the claim in the paper.
> > >
> > > Also, the additional experiment about computational advantages of reparameterized sampling still does not clearly show how the proposed method is superior. As the authors also pointed out, the numbers are not directly comparable as DVGO uses CUDA kernels, and the computational advantage, which is the main contribution of this paper, cannot be demonstrated with this experiment.
> > >
> > > Overall, the paper needs to have more evaluation to demonstrate the computational advantage and more analysis of how the “reparametrized” equation is related to the original equation. I would keep my rating rejecting the paper.

---

### Meta-Review · Area_Chair_w1TM · 2022-08-27

**Recommendation:** Reject
**Confidence:** Certain

**Metareview:**

Reviewers noted that the paper contains some useful ideas for acceleration of neural rendering pipelines.  However, the paper as initially presented was rather preliminary with very limited evaluation.  The authors present considerably more material in the rebuttal, but as noted by the reviewers post-rebuttal, this extra material fails to demonstrate the computational advantage which is a primary contribution of this method.   It is hoped that the authors can continue this line of work, following the reviewers' suggestions, and demonstrate the method's value.   It may be that these ideas have other advantages, or that the advantage is ultimately less significant, in which case a more specific venue such as 3DV may be appropriate.

Although 2v6p did not provide a post-rebuttal comment, their initial review  was closely consistent with the others (albeit marginally more positive), so the post-rebuttal analysis of the reviewers and AC remains valid.


**Award:**

No

---

### Decision · Program_Chairs · 2022-09-14

Reject